# Quinoa Productivity and Stability Evaluation through Varietal and Environmental Interaction

**DOI:** 10.3390/plants10040714

**Published:** 2021-04-07

**Authors:** Elhadji Thiam, Asmaa Allaoui, Ouafae Benlhabib

**Affiliations:** Département de Production, Protection et Biotechnologies Végétales, Institut Agronomique et Vétérinaire Hassan II, Rabat B.P 6202, Morocco; allaouiasmaa81@gmail.com

**Keywords:** quinoa, agro-morphological traits, genotype × environment interaction, AMMI, yield, stability

## Abstract

*Chenopodium quinoa* is a pseudocereal species identified as a potential crop to mitigate world food security. It has the ability to adapt to diverse agro-ecosystems ranging from sea level to over 4000 masl. Its cultivation in Morocco began in 1999, as it is tolerance to drought, salinity, and frost, and it can grow on marginal soils. It has exceptional nutritional value, as it is rich in proteins, essential amino acids, mineral nutrients, trace elements, vitamins, and unsaturated fatty acids. The present study aims to evaluate the adaptation of 14 quinoa varieties and lines from four different origins through fourteen agro-morphological characters. The experimental trials were conducted at five contrasted agro-climatic sites across the central part of Morocco. The data analysis showed high variability among the tested varieties and between sites for all assessed traits. The Meknes (foot-hill plain) site was the most productive; its grain yield reached 78.6 qx/ha. At the Rabat (coastal land) and Berrechid (continental plain) sites, grain production was respectively 56.4 and 45.9 qx/ha. The SW2 Moroccan line produced the highest grain yield that reached 78.3 qx/ha across sites. The Danish variety Titicaca presented the best harvest index (HI = 0.69) as well as the best “thousand kernel weight” (TGW = 3.4 g). As the mildew infection evaluation, the Vikinga and Titicaca varieties ranked the most sensitive to *Peronospora farinosa*. The germination rates of the harvested seeds were prejudiced by the sites’ high temperatures and were low in Tinejdad (oases site) and El Kbab (mountain plateau). The best average germination rate across sites was that of the Puno variety (84.5%). According to the Additive Main effects and Multiplicative Interaction analysis (AMMI), 23% of the grain yield variability is due to the genotype, while 32% is due to the site by the variety interaction contribution to the production variability. AMMI analysis also ranked the varieties according to their productivity and stability value. Accordingly, two varieties that have yielded above the overall average (42.7 qx/ha) are considered stable; those are Riobamba and W11, which is a local selected line. Titicaca, ILLPA, Atlas cultivars and the SW2 local line presented the best grain yield in one of the experimental sites but performed not as well on the others.

## 1. Introduction

Global warming in recent decades reached a critical level exposing some populations at risk, particularly in terms of the food supply. Food security is also threatened by population growth and rainfall decrease. Grain crops statistics predict a yield decrease between 3.1 and 7.4% for every 1 °C temperature increase unless breeders create new varieties for warmer environments [1]. Thus, there is a great challenge to overcome by exploring and expanding new crops that have a better adaptation to the harsh climates.

Quinoa is a pseudocereal native to the Andean highland of the South. It gained worldwide attention because of its tolerance to abiotic stresses [2] such as drought, heat, frost, and salinity [3]. It expends progressively in different parts of the world; it is growing in more than 95 countries [4]. The high resistance to the abiotic stresses in quinoa results from its vast genetic diversity and harsh environmental conditions prevailing in its zone of origin [5]. It sustains five ecotypes based on their adaptation ability to specific agro-ecological environments: sea level from Southern Chile, Andean valleys, Yungas from the subtropical rainforest, Salar from the Lake Titicaca highland in Southern Bolivia, and Altiplano from the Andean high plateaus (ca. 4000 m) [6].

Quinoa is also known for its high nutritional value; its seeds are exceptionally rich in proteins and essential amino acids. Temperature and photoperiod are the two main factors that affect significantly quinoa production [7]. Quinoa yield is generally insignificant in regions where the temperatures go higher than 32 °C [4,8]. According to Bertero et al., [9], the genotype’s performance depends largely on the cultivar genetic makeup, the environment, and their interaction. In general, quinoa shows high genotype × environment interaction (GEI) under multi-environments trials (MET). Such significant GEI affects breeding efficiency strength [10]. Understanding and assessing genotype and GEI effects are required to enhance the selection efficiency in crop breeding [11]. Furthermore, MET has the advantage of helping identify genotypes that have large adaptations or the ones that adjust to a specific environment [12].

Several statistical analyses are used to evaluate the yield performance of genotypes across the environments such as the regression coefficient of the GEI effects on the environmental [13], the coefficient of variation (CV) [14], the non-parametric stability [15], the harmonic mean of the genotype relative performance value [16], and the AMMI—Additive Main effects and Multiplicative Interaction model [17]. There are different crops that benefit from these approaches, including corn [18], rice [19], bread wheat [20], maize [21], and quinoa [22,23]. AMMI analysis allowed us to analyze the GEI effects in multi-location trials. This method has the advantage of generating outputs that can help easily diagnose the varietal adaptability and yield stability [24].

Quinoa was selected among half a dozen species as elite to enhance crop diversity of the cropping systems in Morocco. The quinoa introduction’s main objective was to improve the food security of mountain sustenance farmers in climate change and overgrazing occurrence. Since then, quinoa became the focus of several studies, germplasm productivity, and adaptation evaluation, screening for diseases and abiotic stresses tolerance, water-use efficiency and drought tolerance, genetic and molecular characterization, etc. [25,26,27]. Consequently, quinoa cultivation has expanded across several regions and environments in Morocco; hence, there is a real need to provide well-adapted cultivars to farmers and producers for different regions. Therefore, the main objectives of the present study are to evaluate the productivity and stability of 14 quinoa genotypes (11 varieties and three Moroccan lines) under field conditions at five contrasting environments and to interpret the genotypes adaptation through the tested cropping conditions and through 14 evaluation descriptors.

## 2. Material and Methods

We conducted the experiments at five sites during the 2017–2018 cropping seasons. Table 1 summarizes the specifications of the experimental locations, the quinoa genotypes under evaluation, the type of environmental climates (oceanic to semi-desert), and altitude ranges (low to high) of the sites. The field experiment design was a randomized complete block with four replications. The elementary plot size varies according to the available land space for the trial. Six rows represented the elementary plots in Rabat and El Kbab, four rows represented the elementary plots in Berrechid and Tinejdad, and three rows represented the elementary plots in Meknes. The row length was respectively 1.6, 2.0, or 2.5 m with a 0.2, 0.5, or 0.6 m’s inter-row spacing. All the cropping tasks (planting, weeding, harvesting, and cleaning seeds) were manual. The fourteen genotypes tested are from different origins: Titicaca, Puno, Vikinga (Quinoa Quality, Regstrup, Denmark), Atlas, Pasto, Riobamba (Wageningen University, Wageningen, The Netherland), ILLPA, Amarilla de Marangani, Altiplano INIA 431, Salcedo INIA, Passankalla INIA 415 (Universidad Nacional Agraria de La Molina, Lima, Peru), and SW2, W11, and W16 (IAV Hassan II, Rabat, Morocco). We sowed all the genotypes at a rate of 0.66 g/m in February during the cropping season expect El Kbab, which was sowed in April since the snow.

The mildew (*Peronospora farinosa*) assessment took place under the natural infection conditions at the vegetative and flowering plant growth stages. We first collected three leaves from three stem levels, the third-bottom, the middle, and the apical per genotype and block, and scored them according to the leaf area surface covered by the fungus [28].

At harvest, we scored the agro-morphological traits. We pulled up separately four rooted plants per elementary plot. Subsequently, we assessed seven IBPGR traits (descriptors International Board for Plant Genetic Resources) that are good in discriminating traits between genotypes or well correlated to the yield in our former screening tests [29,30]. These gathered plant height, stem diameter, root length, panicle length and width, and yield components.

Grain and dry matter yield per plant measurements took place after threshing and drying the seeds at 35 °C and brushwood at 70 °C immediately. Three samples of one thousand kernels that were weighted were obtained by counting the seeds manually. For the grain diameter, thirty seeds per genotype and per block were measured using a binocular magnifying glass with mm graph paper. Yield and dry matter per hectare were estimated by through the grain yield per plant, plant density, and elementary plot area. Harvest index was calculated as defined below: HI = (GW/(BW + GW)) × 100)GW: Grain weightBW: Brushwood weight

For the germination test, we held three samples of 20 kernels per treatment (genotype × site), washed them in 12% sodium hypochlorite solution, and then placed them at room temperature (20.2 °C) in sterilized Petri dishes. Daily recording germinating rate data recording was for ten days.

The database went first through the descriptive analyses and ANOVA test to assess the variability between the genotypes within and between sites. Genotypes homogeneous groups were according Newman–Keuls posthoc test. The Principal Component Analysis [31] and the Additive Main effects and Multiplicative Interaction (AMMI) model [17] performed the genotype stability and productivity. The version 3.5.1 of R software version 3.5.1 is the one used in the analyses. The AMMI model combines both the ANOVA and the PCA. The principal additive effects divide the effect of the GEI into interaction principal component axes (IPCA) provided by the PCA. The calculation of the AMMI Stability Value (ASV) is according to Purchase [32], where $\frac{{\mathrm{Sum}\;\mathrm{of}\;\mathrm{Square}}_{\mathrm{IPCA1}}}{{\mathrm{Sum}\;\mathrm{of}\;\mathrm{Square}}_{\mathrm{IPCA2}}}$ is the weight given to the IPCA1 value that is equal to the IPCA1 sum of squares divided by the IPCA2 sum of squares. Genotypes with small values of IPCA1, IPCA2, and ASV are more stable across environments. It also means that the genotypes with the ASV close to zero are the most stable ones [32].

## 3. Results

Analysis of variance showed a high degree of morphologic and agronomical variability among the genotypes. For all the investigated traits, a significant effect exists except for the germination rate and the seed size in Rabat and Tinejdad (Table 2). The plant height fluctuates between sites from 34.85 to 127.35 cm at El Kbab and Tinejdad, respectively. The ANOVA reveals the tall size of the Peruvian varieties; Amarilla de Marangani reached 2.30 m at Rabat. Pasto and Vikinga cultivars produced the smallest size plants. At the Meknes site, the Moroccan lines were taller than some Peruvian varieties. Intermediate sizes over the sites were mainly those of Puno and Titicaca, which are the Danish cultivars.

Plant dry matter varied between sites, from an average of 2.31 to 53.02 g in respectively El Kbab and Tinejdad; while plant grain average yield ranged from 0.81 to 14.22 g at El Kbab and Meknes, respectively (Table 2). Therefore, plants produced more biomass at Tinejdad and less at El Kbab and more seeds in Meknes than El Kbab. Furthermore, Atlas had the highest grain yield per plant in Meknes, averaging 50.75 g; it also had a good performance in Tinejdad and Berrechid, where its plant grain yields reached 32.17 and 21.38 g, respectively. In Berrechid, Titicaca ranked first with a plant grain yield of 25.19 g. In Rabat, W11 had the best grain yield per plant of 33.06 g, which was followed by both other lines, W16 and SW2 (27.63 and 25.63 g). The Peruvian varieties were the least and even did not produce any seeds in Tinejdad.

The harvest index fluctuated significantly between sites from 0.22 to 0.42. The highest harvest index of 0.69 belongs to Titicaca at Meknes. Puno presents the best harvest indexes of 0.64 and 0.65 at Rabat and Berrechid, respectively (Table 2).

The powdery mildew (*Peronospora farinosa*) sensibility evaluation took place in Rabat and Meknes sites. The reaction to the disease was more pronounced at Rabat (26.84%) than Meknes (17.59%). Four varieties were the most susceptible, Vikinga with 45.50 and 58.50% respectively at Meknes and Rabat, Titicaca with 51.75 and 34.00%, SW2 with 37.25 and 40.50%, and W16 with 43.00 and 33.25%. Pasto and Puno were the most tolerant with 0.50 and 13.25%, and 8.25 and 12.50%, respectively, at Meknes and Rabat (Table 2).

After their harvest and conditioning, collected seeds were placed in Petri dishes for germination to test their viability. The site effect was significant on the germination rate. Thus, the data reveal Meknes and Berrechid as the best sites for seed production; their germination rates exceeded 93%. Rabat recorded a germination rate close to 80% (Table 2). El Kbab and Tinejdad germination rates were below 30%. Puno had the best germination percentage across sites (84.5%) with 100% at Meknes and 50% at Tinejdad.

To assess the varietal variation, principal component analysis (PCA) was performed by considering simultaneously all the variables. The first two principal components explain 72% of the total variability between the 14 genotypes under study. Plant height, stem diameter, dry matter per plant, and harvest index were the major contributors to PC1. Thus, PC1 is considered as a biomass production indicator. The second principal component (PC2) was more related to grain yield per plant, root weight, and plant density, which partially explains the seed production ability per surface unit. The PC3 contributes 10.24% to the total variance; root length, panicle width, grain yield, and plant density present the largest coefficients. The PC4 axis links mainly to the mildew sensitivity (52.41%) and the thousand kernel weight (19.34%). The three first components explained 85.49% of the variability. Thus, these axes were useful to identify homogeneous genotype clusters.

According to the Pearson correlations matrix, the variables that contribute more to the biomass are plant height, stem diameter, root and stem weight, panicle length and width, and plant dry matter. They all are positively correlated to the dry matter and negatively correlated to the harvest index and thousand kernel weight. Genotypes with high growth vigor did not produce much seed; that was the case of the Peruvian cultivars. In Rabat, Passankalla, Salcedo, ILLPA, and Altiplano, dry matter ranged between 14.6 and 19.3 g/plant, while in the Danish cultivars, it varied from 3.9 to 5.2 g. At the same time, the Danish genotypes harvest index (0.62 to 0.64) was higher than the Peruvian ones (0.04 to 0.17).

Grain yield, grain yield per plant, harvest index, and thousand kernel weight are the main variables that are positively correlated. Since plant grain yield correlates negatively to the dry matter, cultivars with moderate growth vigor and a short growth cycle seem more productive under the experimental conditions if we consider the February sowing date. Susceptibility to mildew does not correlate with the other traits; rather, it depends on the environment’s climate, mainly temperature and humidity. A strong correlation exists between the plant height and the stem diameter (0.92).

The PCA biplot gathered the 14 genotypes into four clusters (Figure 1). Amarilla de Marangani forms Cluster I (blue) and has high biomass components (plant height, stem diameter, panicle length, and dry matter) as well as a harvest index close to zero. Cluster II (red) includes SW2, W11, W16, and Atlas that have in common relatively low plant density and high grain yield: 63.7 qx/ha on average over the five sites. Cluster III (green) gathers only European cultivars, Titicaca, Puno, Vikinga, Pasto, and Riobamba. This group had relatively low biomass, 7.23 g/plant on average over the five sites. Their cumulative dry matter/plant (36.08 g) does not surpass that of Altiplano (43.15 g). In addition, their harvest index (HI = 0.5) is much higher compared to the other varieties, as is their grain yield average across the sites (49.0 qx/ha). Titicaca seems to have additional special traits than the other cultivars; it has higher yield (72.6 qx/ha), harvest index, and thousand kernel weight, but it is more susceptible to mildew. Cluster IV (yellow) holds Altiplano, Passankalla, Salcedo, and ILLPA Peruvian cultivars; they produced few seeds, an average of 24.5 qx/ha over the five sites; their harvest index is exceedingly low at 0.09 on average, but they are more tolerant to mildew.

In multi-location trials, the selection for grain yield stability involves an estimation of the interaction between the genotypes and the environments (GEI), which is sometimes highly significant. The AMMI analysis on yield facilitates the identification of the most productive and stable genotypes. The analysis of variance (ANOVA) associated with the AMMI model revealed the highly significant sites, genotypes, and interaction effects (Table 3). The environment explains the largest grain yield variability (45.17%) followed by the interaction (31.77%); this indicates the great influence of the selected environments on the cultivars’ behavior.

Furthermore, the first interaction component IPCA1 generated by the model explains 53.4% of the total yield variation, while the second axis justifies an additional 25.4%. Therefore, the IPCA1 vs. grain yield biplot fully describes the quinoa genotypes’ behavior as confirmed through its significance (*p*-value = 0.0001).

Figure 2 illustrates the genotypes grain yield across the five sites. Meknes is the most productive site (78.6 qx/ha), while the lowest yielding site is El Kbab (9.5 qx/ha). Meknes, a propitious pluvial zone with an average annual rainfall of 576 mm/year, mean temperature of 19–22 °C in May–June, and relative humidity of 60–78%, gave much more auspicious growing conditions. The coastal site of Rabat presented a quite high yield of 56.4 qx/ha. The growing conditions of the oasis site of Tinejdad affected significantly the quinoa genotypes’ performance, especially Peruvian cultivars. As a result of the late sowing at El Kbab, the quinoas’ reproduction-phase matches the long photoperiod and high-temperature period (37.2–40 °C), while almost no irrigation is applied. El Kbab conditions impact the 14 quinoa genotypes and more the Peruvian cultivars.

Figure 2 reveals Meknes and Rabat as the most interactive sites with the first IPCA, since they reached the highest IPCA1 scores of 6.88 and −8.54, respectively. These two sites were judged appropriate to discriminate and assess the genotype grain yield variance stability. More than 50% of the genotypes produced over-mean yields; those are W16, Pasto, Riobamba, W11, Puno, Titicaca, Atlas, and SW2. According to Purchase [32], the closer the genotype score is to the center of the IPCA1 by IPCA2 biplot, the more stable is the genotype, and the opposite is true. The highest yield of 78.3 qx/ha was that of SW2. This local accession reveals low stability; it is placed far from the IPCA1–IPCA2 biplot center (Figure 3). Atlas and Titicaca have significant yields of respectively 75.6 and 72.6 qx/ha and are also unstable. The lowest yields were those of Amarilla de Marangani (0 qx/ha), Passankalla (9.2 qx/ha), and Viking (18.6 qx/ha) (Figure 2). Nevertheless, their stability scores were valuable, of 2.46, 1.88, and 3.09, respectively.

Genotypes projection, aside from the environmental vectors, indicated specific interactions. The Figure 3 biplot displays varietal adaptation degree to a specific site. Titicaca is better suited to Berrechid (143.2 qx/ha); ILLPA interacts positively with Meknes environment (125.4 qx/ha), as Atlas (143.9 qx/h) and SW2 (160.4 qx/ha) interact with Rabat, where they recorded their best grain yields. W11 (56.0 qx/ha) and Riobamba (50.7 qx/ha) genotypes, with a yield over the total mean, presented good yield stability.

In summary, the AMMI analysis provides the genotypes’ productivity and the AMMI Stability Values (ASV), as shown in Figure 4. Both parameters allowed gathering the cultivars into three groups.

The first group includes five cultivars, Altiplano, Riobamba, Passankalla, W11, and Amarilla; these are most stable with ASV scores ranging from 1.40 to 2.46. Two of them, Riobamba and W11, are above the average yield (Figure 4).

The second group gathers Vikinga, Puno, W16, and Pasto genotypes; their ASV stability scores range from 3.09 to 4.75; three among them, Puno, W16, and Pasto, overpass the average yield of 42.7 qx/ha.

The third group gathers Titicaca, Salcedo, ILLPA, Atlas, and SW2; their stability scores are sited between 7.75 and 14.78 ASV values. Atlas and SW2 have the highest productivity of 75.7 and 78.3 qx/ha, respectively. Both accessions were ranked unstable according to their ASV scores (12.1 and 14.8, respectively).

## 4. Discussion

Quinoa is an alternative to global stable food products, such as rice and wheat; it is superior in terms of nutritional value and abiotic stress tolerance [33]. Recently, quinoa cultivars have been widely tested outside the species center of origin in Latin America [4]. Quinoa has great morphological, agronomic, and physiological variability, predisposing it to wide environmental adaptation. This great intraspecific diversity connects to the quinoa five ecotypes of different origins and its allotetraploid status [6]. Hence, there is a real need to consider the effect of environmental factors such as temperature and photoperiod to optimize the introduction of quinoa in new regions [34]. On the other hand, when we select genotypes only for their predicted genotypic productivity, it does not guarantee that they will maintain their performance when cultivated under other environments [35]. Thus, it is difficult to identify consistently superior genotypes across environments when the G x E interaction is highly significant [20]. Therefore, this present study seeks to assess the stability of the grain yield of 14 quinoa genotypes across five agro-ecological zones in Morocco.

The data analyses reveal an important agro-morphological diversity between the genotypes. Peruvian cultivars plants have a large size and tend to produce much biomass. In general, their height is within the range reported by Jacobsen and Stolen [36]. According to these authors, quinoa plant size varies between 0.5 and 3 m with an average of 1 to 1.5 m. Amarilla de Marangani gave the highest plant height (2.30 m) as expected; it belongs to the Inter-Andean valleys’ ecotype [37]. Its origin latitudinal score is 13°3 S. According to Tapia et al. [6], this ecotype grows between 2000 and 4000 m, and it has a high size and late flowering. Generally, plant size is much more variable and depends on the plant branching habit and the inflorescence shape, amaranthiform, or glomeriform [6]. The growth cycle duration explains largely plant size differences between the two groups. European varieties were ready for harvest at least 4 weeks before the Peruvian accessions. Peruvian varieties were still at the flowering stage, even Amarilla de Marangani was at the late vegetative phase when the European cultivars reached their physiological maturity. Amarilla de Marangani is the latest and the tallest among all the varieties.

The growth cycle length seems to have a direct relationship with the photoperiod. According to previous studies conducted by Christiansen et al. [38] and Bertero et al. [7], quinoa is a facultative short-day species rather than qualitative, indicating that regardless of the cultivar adaptation, quinoa can flower under a wide range of day lengths. Thus, genotypes are classified as a short day (12.25 h), long day (14 h), or insensitive, depending on their sensibility to the photoperiod. Bertero [34] reported that long-cycle quinoa cultivars are generally more photoperiod sensitive compared to short-cycle ones. Furthermore, the duration of the phenological phases (emergence to floral initiation, floral initiation to the first anthesis, and first anthesis to physiological maturity) depend on the genotype sensitivity to the photoperiod. Christiansen et al. [38] reveal a significant varietal effect on the leaf and flower nodes initiation when quinoa plants are grown at a long day compared to short-day photoperiods, with Real being the most pronounced reactive cultivar. As Real is a traditional large-seed variety from the Bolivian southern Altiplano, it is late-maturing when compared to the sea level ecotype such as the Danish varieties. Our findings match with these statements. Peruvian late cultivars produce more vegetative organs at the detriment of the seeds. In addition, Bendevis [39], while testing two quinoa cultivars, reported similar adjustments in resource allocation between plants growing under short and long-day photoperiods. They are also in agreement with Dorais and others’ [40] conclusions on tomato and sweet pepper crops, stating that an extended photoperiod results in further shoot development and carbohydrate accumulation in leaves over fruit development.

In contrast, when quinoa grows under long photoperiods and high temperatures, it tends to shorten its development cycle. Therefore, plants might reach their physiological maturity even before they complete their development. This should be the case in El Kbab where the plants’ size did not exceed 54 cm. It is important to remember that the El Kbab trial was installed 7 weeks later because of the snow, and there was no irrigation stream when the temperature and day length increased. These results confirm the Atkinson and Porter [41] scheme stating that when the growing environment is at risk of high temperature or water stress, plants would hasten their development and form their seeds. Bertero [34] reported that late in the season in the Andes, when water deficit occurs, photoperiod-sensitive cultivars are faster in their growth and seed filling, allowing plants to mature before water or temperature becomes restrictive.

Water supplies may also influence plant growth to some extent. All the genotypes were relatively more vigorous in Tinejdad despite the oases type of climate (average temperature 23 °C, humidity 33.83%). Indeed, the Tinejdad trail received regular irrigation to meet the plant and the atmosphere demand, which contributed to the expansion of the plant size. In fact, excessive irrigation at the seedling stage improves the plant vigor but does not enhance the grain yield [42].

The origin of the variety appears to impact considerably quinoa seed production. Most Peruvian cultivars produced little or no seeds despite their well-established panicles. They were very late when compared to the other genotypes. Amarilla de Marangani did not produce any seeds across the five experiment sites. Several tested genotypes are day-length sensitive and classified as short day. Quinoas are also sensitive to temperature; they require between 15 and 20 °C for optimal growth. Thus, the growing cycle duration depends on the combination of both factors [43]. Bertero [43] reported that far from the equator, short-day plants become more sensitive to temperature and less sensitive to photoperiod. In 2017, Noulas [44] reported that several Latin American cultivars gave no seeds under the central Greece arid environment. Research conducted also in the US stated that temperatures exceeding 35 °C are likely to cause pollen sterility [45]. These facts well justify our current finding on the Peruvian quinoa in the Tinejdad oases. In addition to the high temperatures, a summer-long photoperiod and low air humidity contribute to the seed establishment hindering. On the other hand, the same Peruvian cultivars behave differently at Meknes, where average temperatures are quite mild (19.3 °C). In terms of comparison, ILLPA seed production per plant at Meknes (11.19 g) over-passed Titicaca (8.81 g), Vikinga (6.44 g), Pasto (8.12 g), and Riobamba (6.25 g).

The two most important determinant factors to grain yield are biomass accumulation and its partitioning into the storage organ, which is evaluated through the harvest index. Puno had the best harvest indexes of 0.64 and 0.65 at Rabat and Berrechid, respectively. As for Meknes, the highest harvest index of 0.69 was Titicaca’s. These well-known quinoa cultivars by farmers recorded the best yield and harvest index at previous experiments. In 2010–2011, Titicaca had the highest harvest index of 0.47 during a trial carried out in the semi-arid region of Bouchane. It is quite low compared to the 2018 exceptional rainy cropping season. Indeed, the 2018 season was particularly favorable for seed production, especially in Meknes. Overall, the varietal harvest indexes ranked between 0.3 and 0.69 for respectively W11 in Tinejdad and Titicaca in Meknes, except for the Peruvian varieties, which experienced problems seed-setting due to heat. The principal component analysis also disclosed the European cultivars group for their high harvest index. Jacobsen [33] reported that quinoa’s perfect variety for seed production in northern Europe is the one that matures early and homogeneously. A growing period shorter than 150 days is advantageous. Quinoa should also have a consistently high seed yield, and it should be short and non-branching to facilitate mechanical harvesting. They used primarily Chilean cultivars that are supposed to be the least sensitive or insensitive to photoperiod for seed filling to achieve these goals [34]. The harvest index represents photosynthetic and translocation seed production capacity. It is also sensitive to agronomic practices and environmental conditions [46]. Rojas et al. [47] reported variation in the quinoa harvest index ranging from 0.06 to 0.87.

Thousand kernel weight (TKW) differences were significant between varieties at Rabat and Berrechid. The TKW means fluctuated from 1.17 g for Salcedo in Berrechid and 3.42 g for Titicaca in Berrechid and W11 in Rabat. The favorable pluvial region of Meknes did not discriminate the varieties through their TKW, as 2018 was an exceptional rainy season, and all the cultivars had reached their best seed weight performance.

Recently, multi-location experiments (MTE) became valuable to assess quinoa genotype adaptation in new regions [9,22,23]. According to Gadisa et al. [20], it is difficult to identify consistently superior genotypes across environments when genotype x environment interactions (GEI) are highly significant. In our present case, the GEI effect explains 31.77% of the total grain yield variation compared to the 23.05% genotype effect. As expected through the AMMI analysis, the environment gathers the largest grain yield variability (45.17%), indicating that grain yield depends highly on the environment. These findings are in line with previous conclusions of other research on quinoa, maize, and bread wheat [22].

AMMI Stability Values (ASVs) of the GEI ranks the studied genotypes into three clusters: stable, moderately stable, and unstable (Figure 4). Among the stable genotypes, two (Riobamba and W11) produced more than the overall mean yield of 42.7 t/ha. The moderate stability group gathers three genotypes that exceed the overall average yield: Puno (56.6 qx/ha), W16 (44.7 qx/ha), and Pasto (46.8 qx/ha). Within the unstable group, Atlas and SW2 genotypes had the highest grain yield of 75.6 and 78.3 qx/ha, respectively. Thus, the grain yield of highly productive genotypes is more likely to be influenced by the growing conditions in different environments. Then, AMMI analysis allowed identifying genotype adaption to a specific site. Titicaca suits better to the Berrechid site, while ILLPA is more productive in Meknes. Atlas and SW2 interact positively with Rabat. According to Mohammadi et al. [48], stability should not be the only parameter for selection; usually, stable genotypes would not necessarily reach the highest yield across the experimental stations. Some authors use the Yield Stability Index as the main selection criterion to rank their germplasm [49,50]. In our case, we used the synthetic graph in Figure 4 to interpret both yield and stability criteria.

For cultivar performance evaluation, we calculated grain yield according to its components. Thus, the estimated yields fit within the range 0.00 qx/ha as a minimum for the Amarilla de Marangani cultivar and 78.3 qx/ha as maximum for the SW2 local advanced line with an overall yield of 42.7 qx/ha. Compared to other countries that have recently introduced quinoa cultivation, grain yields are relatively variable probably because of the large diversity of the cultivars, their origins, and the contrasting environments tested. Dost [51] reported 38.7 qx/ha as the highest yield recorded in Egypt, while in Germany, Präger et al. [52] reached a maximum grain yield of 24.3 qx/ha for the Zeno cultivar and Titicaca produced less in the same experiment. Recently, quinoa grain yield in Egypt revealed a maximum of 30.8 qx/ha under irrigation with the Argentinian Regalona cultivar [53]. The relatively high overall yield of 42.7 qx/ha we obtained through our investigation could have three explanations: the performance of selected cultivars used, the intensive care taken to conduct the trials, and the sampling method based on an individual plant. Moreover, Dost [51] reported a maximum yield of 75.0 qx/ha in Lebanon. Another study investigating the genetic variability of 27 lines originating from different parts of the Andean region and South America conducted in the northern part of India had grain yields varying widely between 4.7 and 60.1 qx/ha [30].

## 5. Conclusions

The present study allowed detecting the great variability among the 14 quinoa genotypes through the morphological and agronomic traits. It also showed that grain yield is more influenced by the environment and the genotype–environment interaction, and it was able to prove the significance and challenge of evaluating the varietal grain yield stability across the different contrasting sites. According to AMMI analysis, Meknes was the most productive site, followed by Rabat and Berrechid. Riobamba cultivar and W11 local advanced line showed broad adaptation with a grain yield above the overall yield. The Titicaca, ILLPA, Atlas, and SW2 genotypes fitted better in a particular environment. Some Peruvian long-grow cycle cultivars performed less in terms of grain yield because of their sensibility to the photoperiod and high temperatures. Other investigation studies should be held at different sowing dates and agro-climatic zones to be able to select for quinoa growers and farmers adapted cultivars to their local environment and to cover their needs in seeds.

## Figures and Tables

**Figure 1 plants-10-00714-f001:**
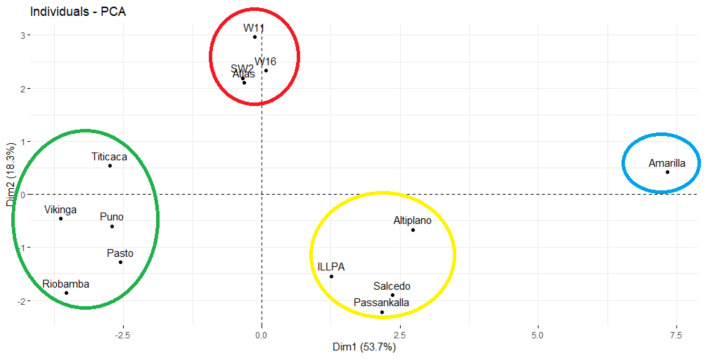
Biplot principal component 1 (PC1) × principal component 2 (PC2) of the 14 quinoa genotypes derived from the average linkage cluster analysis.

**Figure 2 plants-10-00714-f002:**
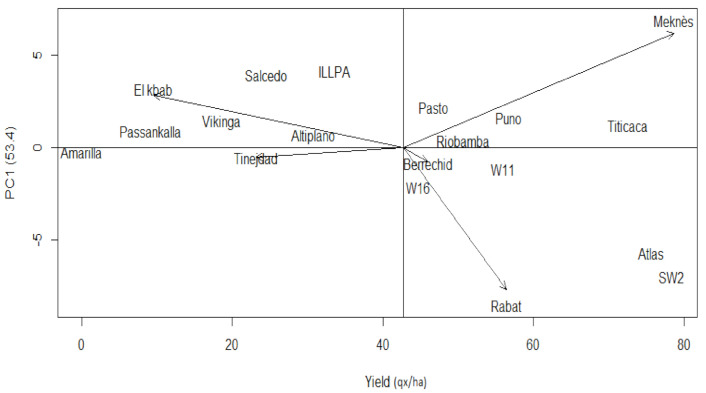
Biplot of the first principal component of the interaction (IPCA1) and the grain yield of the 14 quinoa genotypes.

**Figure 3 plants-10-00714-f003:**
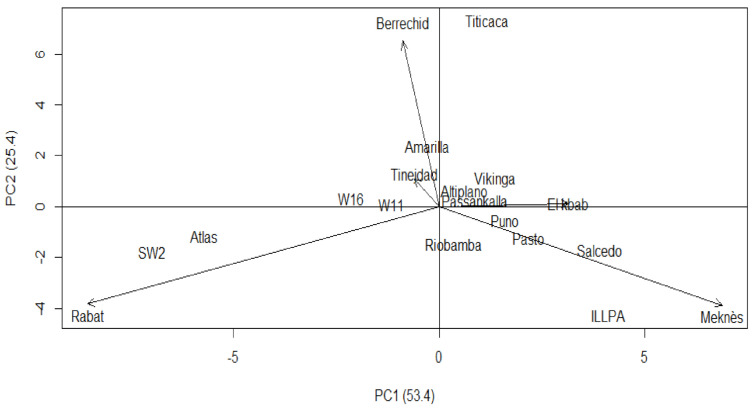
Biplot between the first component (IPCA1) and the second component (IPCA2) of the genotype × environment interaction (GEI) interaction for the grain yield of the 14 quinoa genotypes.

**Figure 4 plants-10-00714-f004:**
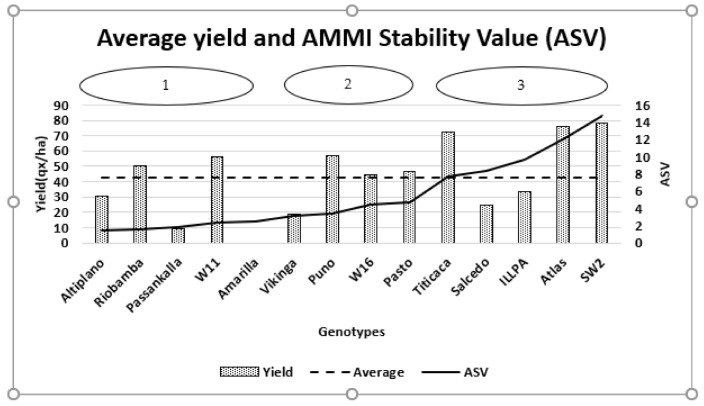
Grain yield and evolution of the AMMI stability values (ASV) of the 14 quinoa genotypes.

**Table 1 plants-10-00714-t001:** Location and description of the experiment agro-climatic sites.

Location	Geographic Position		Alt (m)	Soil Type	Temperature (°C)			Rainfall (mm)	Sowing Date
	Latitude	Longitude			Min	Max	Mean		
Rabat	34°03′31″ N	6°79′10″ W	135	Sandy-Silty	12.6	26.2	19.8	229	20/02/18
Berrechid	33°18′12″ N	7°47′59″ W	309	Clayey-Silty	12	30	20.3	318	16/02/18
Meknès	33°85′42″ N	5°66′17″ W	592	Clayey	11	33	19.3	703	19/02/18
El Kbab	32°71′31″ N	5°55′23″ W	1503	Sandy-Silty	8.2	37.2	19.1	550	06/04/18
Tinejdad	31°54′26″ N	5°19′39″ W	1062	Silty-Clayey	12.1	40.3	23	200	23/02/18

**Table 2 plants-10-00714-t002:** Agro-morphological traits variance analysis of the quinoa genotypes at the five experiment sites.

Sites	Rabat		Berrechid		Meknès		Tinejdad		El Kbab	
Traits	F	Mean	F	Mean	F	Mean	F	Mean	F	Mean
Plant height (cm)	75.623 ***	106.62	23.068 ***	79.33	56.871 ***	97.99	47.938 ***	127.35	3.112 *	34.85
Stem diameter (mm)	13.327 ***	7.67	4.323 ***	8.09	13.543 ***	8.13	19.859 ***	10.97	2.100	4.51
Root length (cm)	2.113 *	15.09	1.249	15.54	11.101 ***	14.77	-	-	1.052	12.56
Panicle length (cm)	14.401 ***	31.73	5.275 ***	24.14	15.831 ***	27.91	-	-	3.765 **	9.15
seed size	NS	1.91	7.639 ***	1.93	28.721 ***	1.99	NS	1.99	33.452 ***	1.72
Panicle width (cm)	7.43 ***	4.74	4.538 ***	4.66	15.831 ***	5.18	-	-	4.373 **	2.03
Dry matter per plant (g)	17.790 ***	18.96	3.688 ***	17.49	9.926 ***	22.29	12.861 ***	53.02	5.935 ***	2.31
Yield per plant (g)	15.711 ***	10.82	5.935 ***	10.14	13.049 ***	14.22	21.652 ***	9.77	7.400 ***	0.81
Harvest Index	59.049 ***	0.4	53.668 ***	0.37	53.136 ***	0.42	26.545 ***	0.22	6.263 ***	0.29
Thousand kernel weight (g)	17.986 ***	2.6	8.323 ***	2.32	1.415	2.85	17.375 ***	2.28	5.682 ***	1.59
Yield (qx/ha)	11.947 ***	56.41	5.541 ***	43.37	3.643 ***	78.64	12.987 ***	9.75	3.310 **	14.92
Dry matter (qx/ha)	15.385 ***	137.91	9.074 ***	132.3	16.796 ***	169.95	20.296 ***	529.39	2.030	383.35
Mildew sensibility (%)	10.196 ***	26.84	-	-	17.540 ***	17.59	-	-	-	-
Germination rate (%)	NS	81.15	NS	93.85	NS	94.62	NS	27.78	NS	29.72

***, **, *: significant at 0.1%, 1%, 5% respectively; NS = Non-significant; Gen = Genotype.

**Table 3 plants-10-00714-t003:** Additive Main effects and Multiplicative Interaction analysis (AMMI) analysis of variance for grain yield of 14 quinoa genotypes in five locations.

	D	Sum Sq.	Mean Sq.	*F* Value	Pr. (>F)	Var. (%)
Environment	4	822,732	205,683	68.3663 ***	9.01 × 10^−10^	45.17
Rep (Env)	15	45,128	3009	2.0224 *	0.01166	-
Genotype	13	419,875	32,298	21.7112 ***	2.20 × 10^−16^	23.05
Interactions	52	578,715	11,129	7.4812 ***	2.20 × 10^−16^	31.77
IPCA1	16	69,269.584	4329.349	2.91 ***	0.0001	53.4
IPCA2	14	32,940.884	2352.9203	1.58	0.0785	25.4
Residuals	983	1,462,333	1488			

***, *: significant at 0.1%, 5% respectively.

## Data Availability

Dataset analyzed in this study is publicly available and can be found here: (https://mega.nz/file/iVc00RxA#Rif61fwxFncPLcNoxz-ciO5sJPxq9cojUlA8dZOX7oE).

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
