# Peer review of "Quinoa Productivity and Stability Evaluation through Varietal and Environmental Interaction"

_plants, 2021, doi:10.3390/plants10040714_

Round 1

Reviewer 1 Report

The manuscript is generally well presented and results are interesting. Authors evaluated quinoa cultivars variability based on genotype and genotype-environment interaction. It's well known that this latter can significally influence the fenotype, but a sistematic study is wellcome in order to drive the farmer's choice of the cultivar based on their local specific pedoclimatic conditions. However, authors should better describe methods, particularlly the chioce of the analysed agronomical and morfological traits and the metod used for recorded them. Generally, agronomical traits of different cultivars are performed according guidelines specific for that specie provided by International Institutions such as IPGRI-FAO.

Best regards

Author Response

Dear,

We are honored to receive your very relevant comments on our manuscript.

We apologize for the delay in coming back to you.

We tried to improve our methodology by providing more details. Also, because the objective of the work is to discriminate for the adaptability and productivity of the varieties, we select those specific descriptors according to former studies.

If there is anything to revise yet let us know.

Thank you for helping to improve our manuscript

Best Regards

Reviewer 2 Report

The whole manuscript lacks in deeper analysis, content organization, and seems simple in content quality. At several places, authors presented fundamental concepts and information with inaccurate scientific explanations without any coherence with each section and title of the article.

Several sections in the manuscript lack experimental evidence as well as important references of authoritative articles. Authors were unable to generate their own synthesis of information in a scientifically correct manner.

In the current version, the review could be improved toward being more balanced, structured, and critical. The authors have to discuss more in functioning photosynthetic apparatus in a changing environment.

The authors analyzed 14 different genotypes and showed that the grain yield of quinoa is more influenced by the environment and the genotype-environment interaction. However, a deeper mechanistic understanding of the plant reactions was not presented and should be improved. Authors could add relevant pieces of information about regulatory mechanisms, plant stress tolerance, and vulnerability to a harmful environment.

The authors have to add the new necessary references to the relevant places.

Hajihashemi S., Skalicky M., et al.: Cross-talk between nitric oxide, hydrogen peroxide and calcium in salt-stressed. Chenopodium quinoa Willd. at seed germination stage. Plant Physiology and Biochemistry, 2020, Vol. 154, 2020, 657-664 doi: https://doi.org/10.1016/j.plaphy.2020.07.022

Jarvis D.E., Ho Y.S., Lightfoot D.J. et al.: The genome of Chenopodium quinoa. Nature, 542 (2017), pp. 307-312, 10.1038/nature21370

Koyro, H.W. et al.: Effect of salinity on composition, viability and germination of seeds of Chenopodium quinoa Willd, Plant Soil, 302 (2008), pp. 79-90, 10.1007/s11104-007-9457-4

In my reading, this MS  deserves publication after revision.

Author Response

Dear,

We are honored to receive your very relevant comments on our manuscript.

We apologize for the delay in coming back to you.

Your comments on the photosynthetic mechanisms of plants are very relevant. But in our investigations, we were specifically interested in discriminating the 14 cultivars through their yield production at several different sites in order to assess their production stability and to be able to recommend some cultivars to farmers. The data we collected did not allow us to go into the physiological details of the plants. This implies that further studies must be carried out in this direction.

The references you sent us on the germination of quinoa in relation to salinity are excellent and will surely serve as a prominent basis for further prospecting on the mechanisms linked to the germination faculty of our germplasm and quinoa in general.

Nevertheless, we reviewed the methodology and the introduction by giving more details as you suggested.

If there are specific parts that need to be reviewed in the manuscript let us know.

Best Regards
